# PD-L1 Expression in Non-Small Cell Lung Cancer Specimens: Association with Clinicopathological Factors and Molecular Alterations

**DOI:** 10.3390/ijms23094517

**Published:** 2022-04-19

**Authors:** Mohammed S. I. Mansour, Karina Malmros, Ulrich Mager, Kajsa Ericson Lindquist, Kim Hejny, Benjamin Holmgren, Tomas Seidal, Annika Dejmek, Katalin Dobra, Maria Planck, Hans Brunnström

**Affiliations:** 1Department of Pathology and Cytology, Halland Hospital Halmstad, SE-301 85 Halmstad, Sweden; kim.hejny@regionhalland.se (K.H.); benjamin.holmgren@regionhalland.se (B.H.); tomas.seidal@regionhalland.se (T.S.); 2Division of Pathology, Department of Clinical Sciences Lund, Lund University, SE-221 00 Lund, Sweden; karina.malmros@med.lu.se (K.M.); kajsa.ericsonlindquist@skane.se (K.E.L.); hans.brunnstrom@med.lu.se (H.B.); 3Division of Respiratory and Internal Medicine, Department of Clinical Medicine, Halland Hospital Halmstad, SE-301 85 Halmstad, Sweden; ulrich.mager@regionhalland.se; 4Department of Genetics and Pathology, Laboratory Medicine Region Skåne, SE-221 85 Lund, Sweden; 5Department of Translational Medicine in Malmö, Lund University, SE-205 02 Malmö, Sweden; annika.dejmek@gmail.com; 6Department of Oncology-Pathology, Karolinska Institute, SE-171 77 Stockholm, Sweden; katalin.dobra@ki.se; 7Department of Clinical Pathology and Cytology, Karolinska University Hospital Huddinge, SE-141 86 Stockholm, Sweden; 8Division of Oncology, Department of Clinical Sciences Lund, Lund University, Medicon Village, SE-223 81 Lund, Sweden; maria.planck@med.lu.se; 9Department of Respiratory Medicine and Allergology, Skåne University Hospital, SE-221 85 Lund, Sweden

**Keywords:** biopsy, cytology, EGFR, histology, KRAS, mucinous, sample site

## Abstract

Immune checkpoint inhibitors (ICI) targeting programmed cell death-1 or its ligand (PD-L1) have improved outcomes in non-small cell lung cancer (NSCLC). High tumor PD-L1 expression, detected by immunohistochemistry (IHC) typically on formalin-fixed paraffin-embedded (FFPE) histological specimens, is linked to better response. Following our previous investigation on PD-L1 in cytological samples, the aim of this study was to further explore the potential impacts of various clinicopathological and molecular factors on PD-L1 expression. Two retrospective NSCLC cohorts of 1131 and 651 specimens, respectively, were investigated for PD-L1 expression (<1%/1–49%/≥50%), sample type, sample site, histological type, and oncogenic driver status. In both cohorts, PD-L1 was positive (≥1%) in 55% of the cases. Adenocarcinomas exhibited lower PD-L1 expression than squamous cell carcinomas (*p* < 0.0001), while there was no difference between sample types, tumor locations, or between the two cohorts in multivariate analysis (all *p* ≥ 0.28). Mutational status correlated significantly with PD-L1 expression (*p* < 0.0001), with the highest expression for *KRAS*-mutated cases, the lowest for *EGFR*-mutated, and the *KRAS/EGFR* wild-type cases in between. There was no difference in PD-L1 levels between different prevalent *KRAS* mutations (all *p* ≥ 0.44), while mucinous *KRAS*-mutated adenocarcinomas exhibited much lower PD-L1 expression than non-mucinous (*p* < 0.0001). Our data indicate that cytological and histological specimens are comparable for PD-L1 evaluation. Given the impact of *KRAS* mutations and the mucinous growth pattern on PD-L1 expression, these factors should be further investigated in studies on ICI response.

## 1. Introduction

Immune checkpoint inhibitors (ICI) targeting programmed cell death-1 (PD-1) or programmed death-ligand 1 (PD-L1) have greatly improved the clinical outcome in non-small cell lung cancer (NSCLC) patients [1,2,3]. Although a variable and moderate therapy response is seen for the whole NSCLC population, better survival rates have been demonstrated for those with high tumor PD-L1 expression [3,4].

PD-L1 expression in tumor cells is currently detected by immunohistochemistry (IHC) mainly on formalin-fixed paraffin-embedded (FFPE) histological specimens [5]. While studies evaluating treatment response in patients with PD-L1 expression based on cytology are rare [6], numerous investigations comparing histological and cytological material for assessment of PD-L1 have demonstrated that cytology could be used in the absence of a biopsy after appropriate quality control [5,7,8].

In our previous investigation of paired biopsies and cytological specimens, we found a good and moderate concordance, respectively, for PD-L1 expression in two independent cohorts [7]. In the former, all discordant cases had lower expression in cytology (with alcohol-based fixative) while in the latter, a lower concordance was seen for pleural effusions than for bronchial cytology [7,9].

Large studies have demonstrated diverging results for correlations of PD-L1 expression to clinicopathological factors and molecular alterations. A systematic review did not find any strong correlations [10], while a later meta-analysis found high PD-L1 expression associated with male gender, smoking, poor tumor differentiation, large tumor size, presence of lymph node metastasis, *EGFR* wild-type status, and *KRAS* mutations (whereas *ALK* translocation status could not be correlated to PD-L1) [11]. A very large study also found that high PD-L1 expression correlates with poor differentiation and *EGFR* wild-type status, but also with cytological sample, sampling from pleural and nodal metastases and *ALK* translocations (no correlation to gender was seen, while *KRAS* could not be fully investigated) [12]. PD-L1 expression in *KRAS*-mutated NSCLC is of great interest as this group constitutes a significant part of ICI treated cases [13,14]. While *KRAS* mutations are common in mucinous lung adenocarcinomas (AC) [15], in our experience, mucinous lung AC are often negative for PD-L1 IHC staining. 

Our previous studies on paired biopsies and cytological specimens were too limited in number for some analyses [7,9]. Therefore, the aim of the present study was to investigate correlations between PD-L1 expression and specimen type, sample site (primary tumor or metastasis), histological type (and specifically mucinous AC), and molecular alterations in larger (nonpaired) cohorts.

## 2. Results

### 2.1. Characteristics of the Cases

During the studied time, molecular analysis was requested for 1275 NSCLC cases in Lund. In 1094 of these, PD-L1 was analysed, which comprised the Lund cohort of the present study. In 37 of the included cases, PD-L1 was stained on both a biopsy and cytological material, which led to a total of 1131 specimens. The Halmstad cohort consisted of 527 NSCLC cases with PD-L1 analysis, where 124 of them had paired cytology and biopsy, i.e., a total of 651 specimens. 

The prevalence of PD-L1 positivity was 55% of the cases in both cohorts at a ≥1% cutoff level. Further details regarding PD-L1 expression using a three-tier scale (˂1%, ≥1–49%, and ≥50%) and relation to sample type, tumor locality, histopathological type, and oncogenic drivers are reported in Table 1. As evident from the table, not all cases were analyzed for all mutations/fusions, due to limited material and differences in panels and routines between the two pathology departments (e.g., regarding *TP53*) and change of methodologies over time (e.g., affecting gene fusion testing). Images of cases with various PD-L1 expression are found in Figure 1.

### 2.2. Correlation between PD-L1 Expression and Clinicopathological and Molecular Features

From the two cohorts, a total of 1381 specimens, where 90 were a cytological sample from a patient with also a biopsy analyzed for PD-L1, with complete data on sample type, tumor location, histopathological diagnosis, and mutation status were used for statistical analyses. The results are presented in Table 2 and Figure 2. 

As evident from Table 2, AC exhibited significantly lower PD-L1 scores than squamous cell carcinomas (SqCC) and “other NSCLC”, while there was no difference between SqCC and “other NSCLC” (Student’s *t*-test, analysis of variance [ANOVA], and chi2 test). Furthermore, the Lund cohort demonstrated significantly but very slightly higher PD-L1 scores than the Halmstad cohort. Comparisons between sample types and tumor location were not significant.

Cases with *EGFR* mutations had significantly lower PD-L1 scores than EGFR wild-types (Student’s *t*-test, *p* < 0.0001), while *KRAS* and *PIK3CA* mutations were linked to higher PD-L1 expression than *KRAS* wild-types (*p* < 0.0001) and *PIK3CA* wild-types (*p* = 0.006), respectively. Cases with *ERBB2*, *BRAF*, or *NRAS* mutations or *ALK* or *ROS1* fusions did not have significantly different PD-L1 scores than those negative for these alterations in corresponding separate analyses (all *p* = 0.12–0.58). As 5 (7%) and 27 (37%) of the 73 cases with *PIK3CA* mutations also had *EGFR* or *KRAS* mutations, respectively, the cases were grouped into *EGFR*-mutated, *KRAS*-mutated and *EGFR/KRAS* wild-type for further analysis. With this subdivision, cases with *EGFR* mutations had significantly lower PD-L1 scores than cases with *KRAS* mutations or neither mutation, while *KRAS*-mutated cases had significantly higher scores than those with no *EGFR/KRAS* mutation (see Table 2). 

The significant difference between histological type (with lower PD-L1 scores for AC than SqCC) and between mutational status (*EGFR* mutation vs. *KRAS* mutation vs. *EGFR/KRAS* negative) remained in multiple regression analysis, as seen in Table 2, while there was no difference between cohorts, sample types, or tumor location. The significances remained if resections and NSCLC other than AC and SqCC were excluded from the analysis. If the cohorts were analyzed independently, the significances remained for the Lund cohort, but only mutational status, not histological type, correlated significantly to PD-L1 expression in the Halmstad cohort (in the Lund cohort, there was only a nonsignificant trend for lower PD-L1 in cytology than biopsies, *p* = 0.08). 

### 2.3. PD-L1 Expression in EGFR-Mutated Lung Cancers

Among the 163 cases with *EGFR* mutations in the two cohorts, 74 (45%) exhibited a deletion in exon 19 while 66 (40%) had a L858R mutation. The PD-L1 expression was similar for the two types, with 39%/9% and 38%/12% exhibiting PD-L1 in ≥1%/≥50% of the tumor cells, respectively (Student’s *t*-test, *p* = 0.91). All four *EGFR*-mutated SqCC were positive for PD-L1, whereof three ≥50%. While almost all small specimens analyzed for PD-L1 expression in the cohorts were sampled as part of the investigational work-up, i.e., before any treatment, there were 14 *EGFR*-mutated cases with sampling after treatment (for investigation of resistance mutations or confirmation of metastasis). In 71%/14% of these ≥1%/≥50% of the tumor cells were positive for PD-L1, which was significantly higher than in those with sampling before treatment (*p* = 0.034). 

### 2.4. PD-L1 Expression in KRAS-Mutated Adenocarcinomas

In the two cohorts, 403 AC harbored a *KRAS* mutation, corresponding to 35% of all 1160 AC cases in the cohorts, or 42% of the 959 AC with *KRAS* analyzed. The most common *KRAS* mutations were p.G12C in 38%, p.G12V in 18%, p.G12D in 15%, and p.G12A in 8% of the *KRAS*-mutated AC. A mucinous growth pattern was seen in 62 (15%) of the 403 AC with a *KRAS* mutation.

PD-L1 expression was seen in 247 of 403 (61%) *KRAS*-mutated AC at a ≥1% cutoff level. Detailed data using the three-tier scale (˂1%, ≥1–49%, and ≥50%) is presented in Table 3 and Figure 2. Using a student’s *t*-test, there was no significant difference in PD-L1 levels between the four most common *KRAS* mutations (*p* = 0.44–0.92), and ANOVA was nonsignificant as well (*p* = 0.88). The PD-L1 levels were significantly lower in mucinous compared to nonmucinous *KRAS*-mutated AC (Student’s *t*-test, *p* < 0.0001).

**Table 3 ijms-23-04517-t003:** PD-L1 expression in *KRAS*-mutated lung adenocarcinomas from the two cohorts combined.

Parameter	All Cases	PD-L1 < 1%	PD-L1 1–49%	PD-L1 ≥ 50%
** *Growth pattern* **				
Non-mucinous	338	105 (31%)	106 (31%)	127 (38%)
Mucinous	62	49 (79%)	10 (16%)	3 (5%)
Undetermined	3	2 (67%)	0 (0%)	1 (33%)
** *Mutation* **				
p.G12C	154	58 (38%)	42 (27%)	54 (35%)
p.G12V	74	29 (39%)	21 (28%)	24 (32%)
p.G12D	61	23 (38%)	20 (33%)	18 (30%)
p.G12A	33	14 (42%)	10 (30%)	9 (27%)
p.G13C	17	7 (41%)	4 (24%)	6 (35%)
p.Q61H	16	7 (44%)	4 (25%)	5 (31%)
p.G13D	12	4 (33%)	4 (33%)	4 (33%)
p.G12S	10	5 (50%)	3 (30%)	2 (20%)
p.G12R	8	3 (38%)	2 (25%)	3 (38%)
p.Q61L	5	1 (20%)	3 (60%)	1 (20%)
p.G12F	4	0 (0%)	2 (50%)	2 (50%)
p.G13V	3	2 (67%)	1 (33%)	0 (0%)
p.A146V	2	2 (100%)	0 (0%)	0 (0%)
p.G21C	1	0 (0%)	0 (0%)	1 (100%)
p.Q61R	1	1 (100%)	0 (0%)	0 (0%)
p.L19F	1	0 (0%)	0 (0%)	1 (100%)
p.Q61K	1	0 (0%)	0 (0%)	1 (100%)

## 3. Discussion

The last decades have seen a paradigm shift in the treatment of NSCLC with the addition of targeted therapy and immunotherapy. This breakthrough has resulted in an increasing number of treatment-guiding biomarkers, including oncogenic driver alterations and PD-L1 expression. Although targeting the PD-1/PD-L1 pathway results in improved survival, not all patients respond well to anti-PD-1/PD-L1 immunotherapy [1,2,3,4]. The optimal role and potential improvement of PD-L1 testing in addition to other mechanisms affecting responsiveness have not yet been fully elucidated. Here, further input from large studies with real-world data may be one way forward. 

Our previous studies on paired cytological samples and biopsies suggested a lower PD-L1 expression in cytology with alcohol-based fixative, and a lower cyto-histological PD-L1 concordance for pleural effusions compared to bronchial cytology [7,9]. In the present study, we wanted to further explore this in larger (unpaired) cohorts, but also include possible correlations of PD-L1 with molecular alterations and pathological characteristics of interest. 

In our cohorts, 55% of the cases were positive for PD-L1 (≥1% positive tumor cells), which is in line with other large real-world studies with patients exclusively or mainly from Europe and North America containing a prevalence of 52–63% [12,16,17,18]. Zheng et al. reported a lower level of PD-L1 expression (43%), in a large study from China [19], while Wang et al. reported a frequency as high as 71% from Canada [20].

In our study, we found only a very slight and not significantly lower frequency of PD-L1 positivity for cytological samples compared to biopsies and for metastases compared to primary tumors. There are few large studies addressing these factors, but the study by Evans et al. instead found a higher PD-L1 expression in cytology [12], while Wang et al. found no difference [20]. Both Evans et al. and Zheng et al. reported a higher PD-L1 expression in metastases [12,19], and in biopsies compared to resections (the latter was in line with our results), but a slightly higher expression in resections has also been reported [17].

In the present study, we found lower PD-L1 expression in AC than SqCC, mainly driven by the Lund cohort, also considering mutational status in multivariate analysis. Lower PD-L1 expression in AC has also been demonstrated in two large studies from China and North America [18,19], while no difference or higher expression in AC has been presented in two other large studies [21,22] and in systematic reviews [10,23].

From our results and the large studies in the literature discussed above, it is difficult to draw strong conclusions about sample type, sample site, and histological type, but there seems to be no consistent support for cytological samples exhibiting significantly different PD-L1 levels than biopsies. In contrast, there is strong support for lower PD-L1 expression in *EGFR*-mutated AC compared to *EGFR* wild-type [11,12,17,18,19,21,24], although there are reports with a nonsignificant difference [20]. Lower tumor mutational burden (TMB), a lower number of tumor-infiltrating lymphocytes (TILs), and poor response to ICI have also been reported for *EGFR*-mutated NSCLC [25,26,27], and any potential role of ICI in cases with *EGFR* mutations is currently discussed [28]. Higher PD-L1 expression may be seen after targeted therapy, at least in cases with therapy resistance [29]. Our data also support higher PD-L1 in *EGFR*-mutated cases with sampling after or during treatment, but a larger number of cases with evaluation of PD-L1 both before and after treatment would be needed for firm conclusions. 

In line with our results, higher PD-L1 expression has been reported for *KRAS*-mutated cases [11,21,22,24], though not always significantly [20]. However, large studies reporting PD-L1 and both *EGFR* and *KRAS* status (particularly with exclusion of *EGFR*-mutated cases, with generally low PD-L1, from the *KRAS* wild-type group) are quite rare. When investigating AC, we found no difference in PD-L1 levels between prevalent *KRAS* mutations, and if the cases would be grouped into mutations, there was no difference in PD-L1 level between KRAS mutations reported to have high (G12A, G12C, G13D, Q61L) or low (G12D, G12R, G12V) Raf affinity [30] (data is not specifically displayed, but is evident from Table 3). However, there was a pronounced difference between mucinous and nonmucinous *KRAS*-mutated AC in our study, something we have not seen reported in the literature. In a retrospective study of mucinous AC, treatment with immunotherapy was associated with better overall survival [31]. Data on *KRAS* mutations and PD-L1 status was not available for all included cases, but considering the positive therapy response of that study and the low frequency of PD-L1 expression of our data, further investigations should address if PD-L1 is a less predictive marker in mucinous than in nonmucinous AC, and both *KRAS* mutation status and mucinous growth pattern should be considered to be reported in studies on ICI response. 

There are some limitations of the present study. Firstly, the retrospective design and the proportion of old archival cases in one of the cohorts should be regarded. This may possibly also have contributed to a higher frequency of AC cases in the group with a cytological diagnosis, though it is unlikely that this has affected the main results. Secondly, we did not have information regarding overall survival, which patients received immunotherapy treatment, and their response to therapy. Apart from its role as a predictive marker, high PD-L1 expression has been inconsistently linked to worse prognosis [10,21,32], and the potential prognostic and predictive value would be possible to evaluate in our cohorts in the future with longer followup time and consideration of stage and treatment. Thirdly, the next-generation sequencing (NGS) panels used during the studied time did not cover some interesting targets, such as *STK11*, *KEAP1*, and *SMARCA4* [33]. Also, many molecular alterations are quite uncommon, and our conclusions are limited to prevalent types. 

## 4. Materials and Methods

The study included two retrospective cohorts from a university-affiliated (Lund) and a regional (Halmstad) pathology department in southern Sweden.

### 4.1. The Lund Cohort

Consecutive NSCLC cases at the Department of Genetics and Pathology, Lund, from January 2018–December 2019 were included based on ordering of molecular analysis. Occasional cases (<1%) during the study period were archival ones from before 2018, analysed upon clinical request. During the covered time, molecular analysis and PD-L1 was requested as part of the clinical workup for all NSCLC cases with adequate material, except that reflex testing was not mandatory for resected tumors but e.g., requested upon recurrence (i.e., applicable to cases without an adequate pre-surgical small specimen). Also, PD-L1 was not always analysed for specimens sampled for analysis of *EGFR*/*ALK* resistance mutations, while complementary ALK and ROS1 analyses were rarely performed for SqCC with an inconclusive first method for detection of fusions (see below). 

The department serves a population of about 1.7 million inhabitants for molecular analysis and PD-L1. About 30–35% were referral cases for predictive analyses with primary diagnostics of the lung cancer samples at a different pathology department. During the study period, rare cases from Halmstad were analysed with NGS in Lund (typically for *RET* fusions as this target was not included in the panel used in Halmstad at the time). These cases were excluded from the Lund cohort in the present study. 

### 4.2. PD-L1 Testing in Lund

The PD-L1 assessment in Lund has been previously described in detail [7]. In brief, biopsied tissue was normally used for PD-L1 testing, but in cases with no or an inadequate biopsy, a cytological specimen was used if available. Also, extra cell blocks were sometimes constructed in the clinical setting for parallel PD-L1 staining during 2017–2019 even if an adequate biopsy existed. 

Cytological material for IHC was fixed in CytoLyt^®^ (Hologic, Marlborough, MA, USA) for a few hours (except for pleural effusions that were only rapidly washed) followed by PreservCyt^®^ for typically 1–3 days before Cellient™ automated cell block preparation (Hologic, Marlborough, MA, USA). Bronchial cytology, including aspirations from lymph nodes, was sometimes mixed in cell blocks to increase the number of tumor cells. 

Regardless of material, PD-L1 was assessed using the 22C3 assay (Agilent/pharmDx, Santa Clara, CA, USA) with staining on a Ventana Benchmark Ultra (Ventana Medical Systems Inc., Tucson, AZ, USA) using the OptiView visualization system. Control tissue (tonsil and placenta) was routinely used on each slide. PD-L1 was evaluated in the clinical diagnostic setting by several different pathologists working with thoracic pathology and PD-L1 daily. 

### 4.3. Molecular Testing in Lund

For histological FFPE specimens, macrodissection was routinely used for analysis of tumor-rich areas. For cytology, May–Grünwald–Giemsa-stained smears were often used for molecular analysis by scraping off and lysing the cells, but cell blocks were also sometimes used. A pathologist who works with thoracic and molecular pathology daily confirmed that only cases with ≥10% tumor cells in the selected material should have proceeded to analysis. 

From March 2018, mutations and fusions were analysed with the Oncomine™ Focus Assay (Thermo Fisher Scientific, Waltham, MA, USA) on an Ion Torrent S5™ (Thermo Fisher Scientific) after extraction using the Qiagen AllPrep kit (Qiagen, Hilden, Germany). Quality and quantity of DNA and RNA was routinely checked (TapeStation, Agilent Technologies, Santa Clara, CA, USA; Qubit™, Thermo Fischer Scientific) before proceeding with analysis. When needed due to insufficient amount of DNA, for example, Therascreen^®^ EGFR RGQ PCR (Qiagen) was used for *EGFR* analysis. Correspondingly, IHC with clone D5F3 (Ventana Medical Systems Inc., Tucson, AZ, USA) and clone D4D6 (Cell Signaling Technologies, Leiden, the Netherlands), respectively, and FISH with Vysis ALK Break Apart FISH Probe and Vysis ROS1 Break Apart FISH Probe, respectively (both Abbott Laboratories, Abbott Park, IL, USA), were used as backup for *ALK* and *ROS1* analysis when needed. 

Prior to March 2018 (i.e., the first two months of the study period), mutations were analysed with the Ion AmpliSeq™ Colon and Lung Panel v2 (Thermo Fisher Scientific, Waltham, MA, USA) with PCR as backup for *EGFR* as above. *ALK* and *ROS1* fusions were analysed with IHC for biopsies and resected material. FISH was instead used for cell blocks, to complement inconclusive IHC results, and to confirm a positive IHC staining. 

### 4.4. The Halmstad Cohort

The study included consecutive NSCLC cases at the Department of Pathology and Cytology, Halland Hospital, Halmstad, from January 2016–September 2021 based on PD-L1 analysis. Almost 20% of the cases were archival ones from before 2016 but stained during the study period upon clinical request or as part of our previous study [7]. During the covered time, molecular analysis and PD-L1 was ordered for all NSCLC cases with adequate material as part of the clinical workup. The department’s reception area for lung samples, PD-L1, and molecular analysis was about 0.32 million inhabitants. The department does not handle surgical lung resections.

### 4.5. PD-L1 Testing in Halmstad

The PD-L1 assessment in Halmstad has been previously described in detail [7]. In brief, biopsied tissue was normally used for PD-L1 testing, but in cases with no or an inadequate biopsy, a cytological specimen was used if available. Cytological material for IHC was fixed in CytoLyt^®^ (Hologic, Marlborough, MA, USA) (except pleural effusions that were only sometimes rapidly washed in CytoLyt^®^) before centrifugation and manual transfer of the material to a cassette within 24 h and subsequent formalin fixation of the cell block. 

All IHC staining was performed with the 28-8 assay (Agilent/pharmDx, Santa Clara, CA, USA) on an Autostainer Link 48 (Agilent/Dako), using the EnVision FLEX visualization system. Control tissue (tonsil, placenta, and either small intestine or appendix) was routinely used on each slide. For each sample, two consecutive sections were stained with the PD-L1 antibody and with Negative Control Reagent, respectively. Also, positive and negative cell lines provided by the manufacturer (Agilent/pharmDx) were included for each run. PD-L1 was evaluated in the clinical diagnostic setting by several different pathologists who work with PD-L1 daily.

### 4.6. Molecular Testing in Halmstad

For histological FFPE specimens, macrodissection was routinely used for analysis of tumor-rich areas. For cytology, cell blocks were typically used for molecular analysis. A pathologist who works with molecular pathology daily confirmed that only cases with ≥10% tumor cells in the selected material proceeded to analysis. 

From February 2017, mutations were analysed with the TruSight Tumor 15 panel (Illumina, San Diego, CA, USA) on an Illumina MiSeq™ (Illumina) after extraction using the Qiagen DNA kit (Qiagen, Hilden, Germany). The quality and quantity of DNA was routinely checked (LabChip^®^, PerkinElmer, Waltham, MA, USA; Qubit™, Thermo Fischer Scientific, Waltham, MA, USA) before proceeding with analysis. For cases with an insufficient amount of DNA for NGS, the EGFR Mutation Analysis Kit (EntroGen, Woodland Hills, CA, USA) was used instead. Prior to February 2017, *EGFR* mutations were detected using the EGFR Mutation Analysis Kit (EntroGen). On request, *KRAS* analysis was performed in *EGFR*-negative cases with the Therascreen^®^ KRAS Pyro Kit (Qiagen) or the KRAS Mutation Analysis Kit for Real-Time PCR (EntroGen). 

*ALK* rearrangements were analysed with IHC (clone 5A4, Novocastra™, Leica Biosystems, Newcastle upon Tyne, UK) and cases determined positive for ALK IHC were confirmed with FISH (Vysis ALK Break Apart FISH Probe, Abbott Laboratories, Abbott Park, IL, USA). Starting in 2018, *ROS1* rearrangements were analysed with both IHC (clone D4D6, Cell Signalling Technologies, Leiden, The Netherlands) and FISH (Vysis ROS1 Break Apart FISH Probe, Abbott Laboratories).

### 4.7. Data Collection and Statistical Analysis

Data for PD-L1 expression (<1%, 1–49%, or ≥50% positive tumor cells), specimen type used for PD-L1 assessment, histopathological diagnosis, and molecular genetic alterations were retrieved from the pathology reports. Slides were reviewed if any data was unclear, and all slides were reviewed for all *KRAS*-mutated AC for determination of growth pattern (nonmucinous vs. mucinous). Student’s *t*-test, chi2, and, when applicable, one-way ANOVA were performed to investigate correlations between PD-L1 expression and specimen type, tumor locality, histopathological diagnosis, mutational status, and growth pattern in *KRAS*-mutated AC. The results remained the same if using Mann–Whitney and Kruskal–Wallis tests instead. In multivariate regression analysis specimen type (biopsies vs. cytologies vs. resections), tumor locality (primary vs. metastasis; unclear/mixed excluded), histopathological diagnosis (AC vs. SqCC vs. other NSCLC), *EGFR* and *KRAS* mutation (*EGFR* vs. *KRAS* vs. negative/other driver; unknown status excluded), and cohort (Lund vs. Halmstad) were included. In this analysis, biopsy and cytological specimen from a single case were included when both existed and had been stained for PD-L1. All *p*-values were determined using two-sided tests, and a *p*-value of <0.05 was considered significant. The analyses were performed with MedCalc version 14.12.0 (MedCalc Software bvba, Ostend, Belgium).

## 5. Conclusions

In conclusion, we did not find any support that PD-L1 expression is significantly lower in cytological samples compared to biopsies, while PD-L1 seems to be lower in AC than SqCC. Furthermore, the expression is obviously lower in *EGFR*-mutated cases, while *KRAS*-mutated AC exhibit higher PD-L1 expression than *EGFR/KRAS* wild-types. There seems to be no difference in PD-L1 between prevalent *KRAS* mutations, but mucinous morphology is linked to lower PD-L1 expression levels in *KRAS*-mutated AC. 

## Figures and Tables

**Figure 1 ijms-23-04517-f001:**
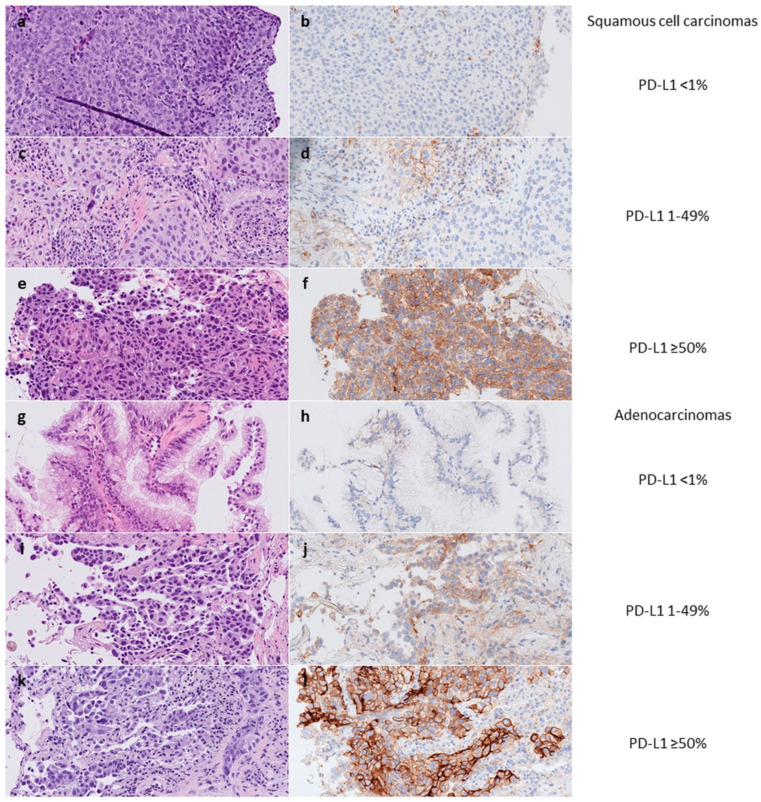
PD-L1 expression in <1% (**b**,**h**), 1-49% (**d**,**j**), and ≥50% (**f**,**l**) of tumor cells in biopsies with squamous cell carcinomas (**a**–**f**) and adenocarcinomas (**g**–**l**). Staining with hematoxylin-eosin and PD-L1 clone 22C3. Original magnification x40 objective.

**Figure 2 ijms-23-04517-f002:**
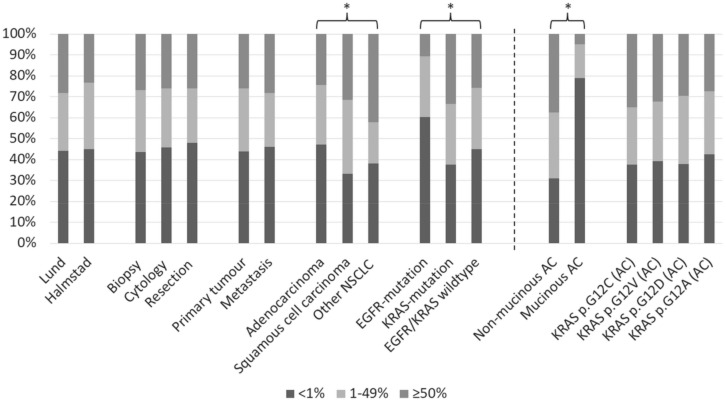
PD-L1 expression levels in all 1381 non-small cell lung cancer (NSCLC) specimens with complete data (left of dashed line) and the 403 KRAS-mutated adenocarcinomas (AC) (* marks statistical significance, all *p* < 0.001; see Table 2 and Table 3 and text for full data and analyses).

**Table 1 ijms-23-04517-t001:** PD-L1 expression and relation to various parameters in two cohorts of non-small cell lung cancer (NSCLC).

Parameter	Lund Cohort	PD-L1 <1%	PD-L1 1–49%	PD-L1 ≥50%	Halmstad Cohort	PD-L1 <1%	PD-L1 1–49%	PD-L1 ≥50%
All cases	1094	492 (45%)	303 (28%)	299 (27%)	527	239 (45%)	174 (33%)	114 (22%)
** *Sample type ** **								
Biopsy	845	364 (43%)	238 (28%)	243 (29%)	469	212 (45%)	160 (35%)	97 (21%)
Cytology	176	85 (48%)	50 (28%)	41 (23%)	180	86 (48%)	46 (26%)	48 (27%)
Resection	110	55 (50%)	29 (26%)	26 (24%)	2 **	0 (0%)	1 (50%)	1 (50%)
** *Locality* **								
Primary tumor	785	340 (43%)	226 (29%)	219 (28%)	469	211 (45%)	161 (34%)	97 (21%)
Metastasis	305	151 (50%)	75 (25%)	79 (26%)	58	28 (48%)	13 (22%)	17 (29%)
Unclear/mixed	4	1 (25%)	2 (50%)	1 (25%)	0			
** *Diagnosis* **								
Adenocarcinoma	776	380 (49%)	208 (27%)	188 (24%)	384	173 (45%)	121 (32%)	90 (23%)
Squamous cell carcinoma	237	80 (34%)	79 (33%)	78 (33%)	137	63 (46%)	51 (37%)	23 (18%)
Sarcomatoid carcinoma/features	15	4 (27%)	2 (13%)	9 (60%)	0			
Adenosquamous carcinoma	3	2 (67%)	1 (33%)	0 (0%)	0			
NSCLC not otherwise specified	63	26 (41%)	13 (21%)	24 (38%)	6	3 (50%)	2 (33%)	1 (17%)
** *Mutations **** **								
EGFR	109 (of 1075)	68 (62%)	28 (26%)	13 (12%)	54 (of 387)	32 (59%)	18 (33%)	4 (7%)
KRAS	304 (of 901)	120 (39%)	79 (26%)	105 (34%)	139 (of 354)	47 (34%)	46 (33%)	46 (33%)
NRAS	7 (of 901)	3 (43%)	3 (43%)	1 (14%)	4 (of 289)	0 (0%)	0 (0%)	4 (100%)
BRAF (V600)	11 (of 901)	3 (27%)	2 (18%)	6 (55%)	6 (of 289)	4 (67%)	1 (17%)	1 (17%)
ERBB2	16 (of 901)	8 (50%)	4 (25%)	4 (25%)	4 (of 289)	3 (75%)	1 (25%)	0 (0%)
PIK3CA	65 (of 901)	19 (29%)	20 (31%)	26 (40%)	8 (of 289)	3 (38%)	3 (38%)	2 (25%)
MET exon 14	13	4 (31%)	3 (23%)	6 (46%)	not analyzed			
TP53	not analyzed				100 (of 289)	45 (45%)	29 (29%)	26 (26%)
** *Fusions **** **								
ALK	20 (of 981)	6 (30%)	7 (35%)	7 (35%)	9 (of 369)	3 (33%)	4 (44%)	2 (22%)
ROS1	1 (of 976)	1 (100%)	0 (0%)	0 (0%)	4 (of 239)	1 (25%)	1 (25%)	2 (50%)
RET	4	1 (25%)	3 (75%)	0 (0%)	not analyzed			
NTRK	1	0 (0%)	1 (100%)	0 (0%)	not analyzed			

* For sample type, results from both a biopsy and a cytological specimen were included if both existed for a single patient (for all other parameters, the PD-L1 result from the biopsy was used if both biopsy and cytology existed). ** Halmstad cases surgically treated in Lund (not included in the Lund cohort). *** Number of tested cases in parenthesis (not recorded for *MET*, *RET*, *NTRK*).

**Table 2 ijms-23-04517-t002:** Correlation of factors to PD-L1 expression in the 1381 lung cancer specimens with complete data.

Characteristics	PD-L1 <1%	PD-L1 1–49%	PD-L1 ≥50%	Student’s *t*-Test	Chi2	Multiple Regression (Coefficient)
** *Cohort* **				*p* = 0.010	*p* = 0.12	*p* = 0.81 (−0.012)
Lund	419 (44%)	265 (28%)	267 (28%)			
Halmstad	193 (45%)	137 (32%)	100 (23%)			
** *Sample type* **				*p* = 0.75 *	*p* = 0.88	*p* = 0.28 (−0.041)
Biopsy	446 (44%)	304 (30%)	274 (27%)	*p* = 0.61 vs. cytology		
Cytology	116 (46%)	71 (28%)	66 (26%)	*p* = 0.81 vs. resection		
Resection	50 (48%)	27 (26%)	27 (26%)	*p* = 0.53 vs. biopsy		
** *Locality* **				*p* = 0.98	*p* = 0.30	*p* = 0.37 (0.049)
Primary tumor	452 (44%)	312 (30%)	269 (26%)			
Metastasis	160 (46%)	90 (26%)	98 (28%)			
** *Diagnosis* **				*p* < 0.001 *	*p* < 0.0001	*p* < 0.0001 (0.17)
Adenocarcinoma	505 (47%)	304 (28%)	261 (24%)	*p* = 0.0003 vs. squamous		
Squamous cell carcinoma	78 (33%)	83 (35%)	74 (31%)	*p* = 0.61 vs. “other”		
Other NSCLC	29 (38%)	15 (20%)	32 (42%)	*p* = 0.0062 vs. adeno		
** *Molecular profile* **				*p* < 0.001 *	*p* < 0.0001	*p* < 0.0001 (0.22)
EGFR-mutation	106 (60%)	51 (29%)	19 (11%)	*p* < 0.0001 vs. KRAS		
KRAS-mutation	179 (38%)	138 (29%)	160 (34%)	*p* = 0.0020 vs. wild-type		
EGFR/KRAS wild-type	327 (45%)	213 (29%)	188 (26%)	*p* < 0.0001 vs. EGFR		

* One-way analysis of variance (ANOVA).

## Data Availability

The data supporting reported results are available from the authors upon reasonable request.

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
