# Peer review of "PD-L1 Expression in Non-Small Cell Lung Cancer Specimens: Association with Clinicopathological Factors and Molecular Alterations"

_ijms, 2022, doi:10.3390/ijms23094517_

Round 1
Reviewer 1 Report
The authors describe a study on PD-L1 expression in bigger (non-paired) cohorts of patients with non-small cell lung cancer. A total of 1381 specimens were used for pairwise Mann-Whitney U tests and multivariate regression analysis, 90 of which were cytological samples from patients who also had a biopsy analyzed for PD-L1, with complete data on sample type, tumor location, histopathological diagnosis, and mutation status.
Major Comments: -
- There are no details on metastases, death, or follow-up for either cohort. It will be interesting to see how PD-L1 expression correlates with metastasis and PD-L1 expression correlates with mutation status in terms of overall survival and metastasis-free survival.
- There are no figures illustrating the various immunoscoring levels of PD-L1 in various types of NSCLC.
- Authors must present all statistical analyses in the form of graphs or tables that are easily understood by the reader.
- In the M&M section, immunohistochemistry and sequencing should be presented in detail with appropriate subheadings.
Minor comment: - Authors must maintain a constant font size.
Reviewer 2 Report
The authors verify the expression of PDL1 and mutation load of KRAS and EGFR. In general the manuscript is well-written.
The major concern I have is that I wonder why authors did not perform correlation statistics between PDL1 and EGFR mutation status and PDL1 expression and KRAS. The authors should report correlation of coefficient statistics.
Second, this is a retrospective study, so could authors perform survival analysis with PDL1 expression in various cohorts.
Round 2
Reviewer 2 Report
No further comments